# Bottle-grade polyethylene furanoate from ring-opening polymerisation of cyclic oligomers

Jan-Georg Rosenboom [1], Diana Kay Hohl [1], Peter Fleckenstein [1], Giuseppe Storti [1] & Massimo Morbidelli [1]

Polyethylene furanoate (PEF) represents a promising renewable resource-based bioplastic as replacement for fossil-based polyethylene terephthalate (PET) with improved material properties. However, the synthesis of PEF through conventional polycondensation remains challenging, since the time-intensive reaction leads to degradation and undesired discolouration of the product. Here we show the successful rapid synthesis of bottle-grade PEF via ring-opening polymerisation (ROP) from cyclic PEF oligomers within minutes, thereby avoiding degradation and discolouration. The melting point of such mixture of cyclic oligomers lies around 370 °C, well above the degradation temperature of PEF (~329 °C). This challenge can be overcome, exploiting the self-plasticising effect of the forming polymer itself (which melts around 220 °C) by initiation in the presence of a high boiling, yet removable, and inert liquid plasticiser. This concept yields polymer grades required for bottle applications ($M_n > 30$ kg mol$^{-1}$, conversion > 95%, colour-free products), and can be extended to other diffusion-limited polymer systems.

[1] Institute for Chemical and Bioengineering, ETH Zurich, Vladimir-Prelog-Weg 1-5/10, 8093 Zurich, Switzerland. Correspondence and requests for materials should be addressed to M.M. (email: massimo.morbidelli@chem.ethz.ch)

While about 40 years ago the decline of oil was a major concern, technological progress has made future shortages of oil a less significant driver for current efforts towards renewable energy and materials[1]. Besides their reduced carbon footprint, new sustainable materials therefore also have to prove their superiority as well as their cost competitiveness with existing fossil-based materials[2]. Polyethylene furanoate (PEF; Fig. 1a) represents a promising 100% renewables-based alternative to its fossil-based counterpart polyethylene terephthalate (PET; Fig. 1a), which is one of the world's most dominant polymers on the market with a production of more than 50 megatons per year[3–7]. However, commercially relevant PEF has been synthesised only via polycondensation, which is a process generally burdened with the necessity of removing the condensation by-products to achieve the polymer chain lengths (i.e., molecular weights) required for the desired material properties. Since the reacting polymer mixture is typically very viscous or even solid, the significantly reduced diffusion of such by-products leads to reaction times of the order of days. The resulting long exposure to the high processing temperatures (around 200 °C) not only increases the production costs, but even more importantly leads to thermal degradation and discolouration of the polymer, which make it unsuitable for the market[8–13]. While for PET the establishment of polycondensation-based processes was more feasible, PEF appears less stable to thermal impact which makes the production of a marketable polymer through such intensive process very challenging[9,10,14].

In order to overcome this issue, we exploit ring-opening polymerisation (ROP)[15] as a faster synthetic route for PEF to deliver sufficient molecular weight on significantly reduced time scales. In this process, the esterification by-products are removed prior to the actual polymerisation by forming macrocyclic oligomers in a diluted non-viscous environment. Since those macrocycles have no end-groups and low ring strain, ROP does not require any by-product removal and is entropically driven. Another advantage is the ability to run controlled and living-like polymerisations, where polymer chains grow simultaneously without termination and molecular weight can be controlled precisely by the amount of initiator. Recent work from our group[16] and Morales-Huerta et al.[17] have shown the feasibility of low to medium molecular weight furanic polyester synthesis via ROP. However, bottle-grade PEF, in terms of absolute molecular weights, conversion and colour, has not been achieved yet.

Here we show that purified cyclic oligomers can be polymerised successfully within minutes below the degradation temperature of PEF, despite the fact that the cyclic oligomer mixture exhibits melting points well above the latter. The application of a cyclic stannoxane catalyst together with effective plasticisation of the system enable the production of polymer grades required for the so-called green bottle ($M_n > 30 \, \mathrm{kg \, mol^{-1}}$, conversion > 95%, colour-free products).

## Results

**Cyclic oligomer synthesis and characterisation**. The synthesis path from the biobased monomers[2] furandicarboxylic acid dimethylester (MeFDCA) and ethylene glycol (EG) to PEF via ROP is outlined in Fig. 1. The cyclic PEF oligomers (cyOEF) can be prepared efficiently by depolymerisation[18,19] of low molecular weight linear PEF oligomers (linOEF) over 6–8 h diluted in high boiling solvents such as 2-methylnaphthalene or 1,2-dichlorobenzene in yields of >95% (Supplementary Figs. 1 and 2). The cyOEF were obtained as a white powder by cooling precipitation and if needed further purified from residual linOEF to 99% purity via silica gel adsorption.

**Fig. 1** Polyethylene furanoate (PEF) and its synthesis via ring-opening polymerisation (ROP). The 100% renewable resource-based monomers dimethyl furandicarboxylate (MeFDCA) and ethylene glycol (EG) are prepolymerised into short linear PEF oligomers (linOEF), which depolymerise in an equilibrium shift towards cyclic oligomers (cyOEF) under dilution in a high boiling solvent. Purified cyOEF can then be polymerised via catalytic ring-opening polymerisation (ROP) within less than 30 min with appropriate plasticisation and initiator to high molecular weight bottle-grade PEF. The comparison of the chemical structures of PEF and PET outlines their chemical similarity but also their distinct difference in aromaticity, which is responsible for various differences in molecular geometry, thermal stability, gas diffusivity and mechanical properties

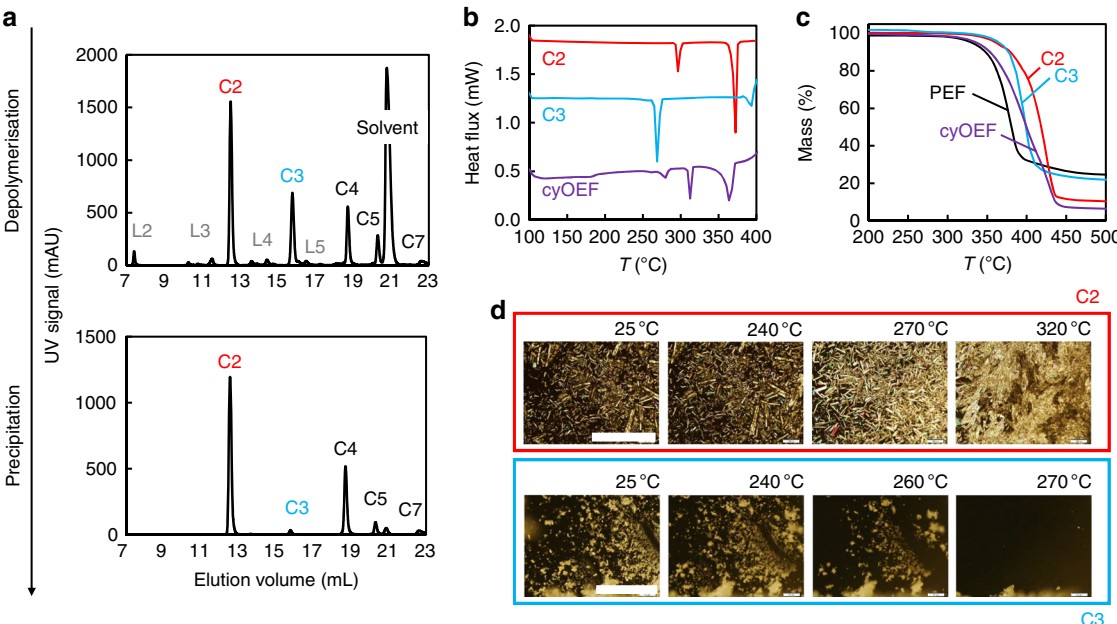

**Fig. 2** Thermal behaviour of cyclic oligomers as raw material for the synthesis of PEF. **a** High-performance liquid chromatography (HPLC) traces of the depolymerisation product show the distribution of cyclic and linear oligomeric PEF species before and after precipitation and purification. **b** Differential scanning calorimetry (DSC) traces reveal the endothermal transitions for isolated cyclic dimer (C2, red), trimer (C3, turquoise) and the typical C2-dominated mixture of cyclic oligomers obtained from depolymerisation (purple), where melting of C2 does not occur until 370 °C after recrystallisation around 290 °C. **c** Thermogravimetric analysis (TGA) weight loss curves indicate thermal degradation (5% loss) of the isolated cycles around 370 °C and the cyclic mixture as well as PEF itself (black) around 330 °C. **d** Polarised microscopy images present the visual proof of C2 recrystallisation at 290 °C observed earlier in the DSC endotherms, while those images for C3 confirm the single melting endotherm at 270 °C. The white scale bars represent 400 μm for the upper row and 200 μm for the lower row of images, respectively

We find that the cyOEF product features a ring-size distribution dominated by the smallest species, the cyclic dimer C2, followed by lower amounts of trimer C3 and larger ring-sizes, as shown in Fig. 2a. While it follows the typical Jacobson–Stockmayer[20] distribution, it is surprisingly different to the reported ring-size distribution of PET that is dominated by the cyclic trimer[19], which can be explained with the difference in thermodynamic ring strain due to the different carboxyl bond angles[21,22]. Moreover, C2 precipitates almost selectively upon cooling after reaction, while the cyclic trimer remains in solution and can only be recovered with an anti-solvent such as hexane. This very specific precipitation behaviour outlined in Fig. 2a was exploited to isolate different mixtures of cyclic species to study their individual characteristics.

The understanding of cyOEF as the raw material for ROP is essential: we find that the ring-size distribution complicates the traditional approach of polymerisation in melt by their strikingly heterogeneous melting behaviour. As observed from differential scanning calorimetry (DSC), C3 melts at 270 °C, while the most abundant species, C2, does not melt until 370 °C (Fig. 2b). Instead, C2 reveals another endothermal transition at 290 °C, where long needle formation observed under the polarised microscope (Fig. 2d) and different powder diffraction patterns (Supplementary Fig. 3) prove the occurrence of a recrystallisation event. The material retains its white crystalline appearance until actual melting occurs above 350 °C, and exhibits strong discolouration shortly after, as observed by opening the DSC samples after heating to different temperatures. These results are congruent with thermogravimetric analysis (TGA) results presented in Fig. 2c, and indicate that the degradation of the typical cyOEF product occurs very close to its melting point, which lies well above the degradation temperature of PEF itself (~329 °C, at 5% weight loss in TGA).

**High molecular weight PEF synthesis via plasticised ROP**. The attempt to polymerise the typically obtained cyOEF (62% C2, 2% C3, 27% C4, 5% C5, 4% C7) below the degradation temperature of PEF at 280 °C only leads to very slow ROP initiated by residual alcoholic and catalyst species, where the low melting cyclics (C3, C4 and larger) react to form PEF within 1 h, while C2 remains almost unaffected in a very heterogeneous system. Complete conversion of C2 takes more than 10 h, at which point polymer degradation has reduced the molecular weight substantially, as shown in Fig. 3a and Table 1 (entry 1), and the obtained product is almost black. In general, cyOEF conversions below 95% are unacceptable for bottle-grade PEF, because not only material use should be economically maximised, but also large amounts of residual cyOEF in the final polymer product may alter the material properties. Therefore, increasing the reaction speed by addition of fresh catalyst is required, which is usually added into the molten monomer to facilitate close contact between these reagents. The catalyst used in this study is a solid powder of cyclic stannoxane[23] (Supplementary Fig. 4), which is known as one of the most promising ROP initiators and also delivered the best results in terms of molecular weight and conversion speed in preliminary studies compared with traditional tin octoate (Supplementary Table 1). In the present case, where the major portion of the raw material (C2) is resistant to melting at applicable temperatures, mechanical grinding is the usual method of choice to enable intimate contact between solid reagents. In fact, 0.1% cyclic stannoxane ground into the cyOEF before reaction at 280 °C increases reaction speed substantially; however, the reaction mixture is still heterogeneous and reaction time long enough to allow degradation and discolouration of the product (Fig. 3b and Table 1, entry 2).

In order to facilitate a pseudo-melting of the cyOEF to make it react to bottle-grade PEF well below its degradation temperature,

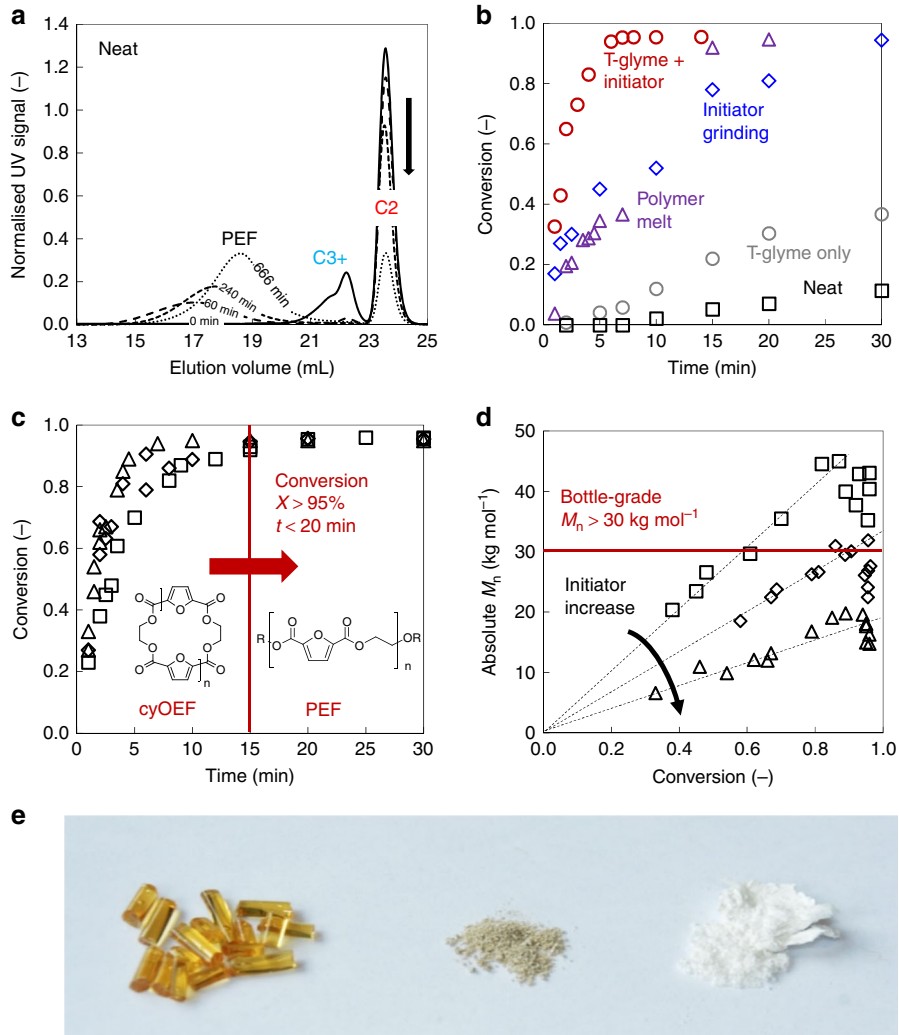

**Fig. 3** Bottle-grade synthesis of PEF via ROP from cyclic oligomers. **a** Size exclusion chromatography (SEC) trace, showing that neat ring-opening polymerisation of 99% cyOEF without additional catalyst at 280 °C leads to relatively fast conversion of C3 and larger cycles to PEF, while C2 polymerisation is very slow and incomplete (black arrow). Simultaneously, the resulting heterogeneous system shows strong discolouration and molecular weight degradation, observable by the shift of the PEF peak to larger elution volumes over time. **b** Comparison of different ROP conditions to enable fast kinetics with short reaction times, all at 280 °C: neat (black squares), dry grinding of 0.1% (mol/mol) cyclic stannoxane initiator into cyOEF (blue diamonds), addition of fresh cyOEF into twice its mass of a reactive PEF melt at >90% conversion with the same initiator amount (purple triangles), 0.1% cyclic stannoxane added together with 33% (mass per mass cyOEF) tetraglyme as liquid inert plasticiser (red circles). The plasticiser alone without initiator (grey circles) increases kinetics to a smaller extent. **c**, **d** ROP using 0.1% (squares), 0.2% (diamonds) and 0.3% (triangles) cyclic stannoxane with 33% tetraglyme at 260 °C, resulting in full ( >96%) cyOEF conversions to high molecular weights exceeding the bottle-grade target of 30 kg mol$^{-1}$ within <20 min. **e** PEF from different routes: a high molecular weight product without undesired discolouration derived from ROP using 0.1% cyclic stannoxane initiator with 33% tetraglyme at 260 °C after 25 min (right), and a discoloured product from non-optimised ROP using 97% pure cyOEF at 280 °C with the same initiator for 60 min (middle), are compared with a typical industrial polycondensation PEF product ($M_n = 15$ kg mol$^{-1}$), which is similarly discoloured (left)

plasticisation is required. We find that the polymer itself has a plasticisation effect due to its much lower melting point (~220 °C) than the cyclic oligomers: pure cyOEF added to a reactive PEF melt (conversion >90%) containing 0.1% initiator is converted to polymer within 20 min at 280 °C (Fig. 3b and Table 1, entry 3). However, no polymer is present at the beginning of the reaction, and therefore reagent mobility is very limited. The idea of using tetraethylene glycol dimethyl ether, commercially also known as tetraglyme, as a temperature-resistant yet volatile liquid and inert plasticiser was then fruitful to overcome this hurdle (Table 1, entry 4). As confirmed by refractive index traces (Supplementary Fig. 5), tetraglyme is present in the beginning of the reaction where plasticisation is required, and leaves the mixture by

evaporation (boiling point = 275 °C) during the later stage, when enough polymer has been formed. An optimised load of 33 wt% tetraglyme at temperatures between 260 and 280 °C is sufficient to obtain a homogeneous reaction mixture and bring the reaction to completion (conversion >95%) within less than 30 min.

Such rapid ROP then allows for tuning the PEF product towards bottle-grade molecular weight and colour. Since plasticisation is essential for a fast reaction to avoid degradation and discolouration, the reaction temperature also plays a key role in this respect: relatively high temperature, e.g. 280 °C, means a fast reaction, but also higher impact of degradation, while polymerisation at 240 °C is insufficient to deliver a fast reaction before degradation becomes significant. ROP at 260 °C results in a

**Table 1 Selected results of ROP for the synthesis of PEF**

| No. | cyOEF purity | Plasticisation | Initiator content | $T$ (°C) | Time (min) | Conv. (%) | $M_n$ (kg mol$^{-1}$) | $M_w$ (kg mol$^{-1}$) | Colour |
|---|---|---|---|---|---|---|---|---|---|
| Effect of plasticisation | | | | | | | | | |
| 1 | 99% | – | – | 280 | 666 | 79 | 14.5 | 25.8 | Black |
| 2 | 99% | Grinding | 0.10% | 280 | 30 | 94 | 33.7 | 67.9 | Beige |
| 3 | 99% | Polymer melt | 0.10% | 280 | 20 | 95 | 33.6 | 63.1 | Beige |
| 4 | 99% | 33% tg. | 0.10% | 280 | 14 | 96 | 31.0 | 61.4 | Beige |
| Effect of temperature | | | | | | | | | |
| 5 | 99% | 33% tg. | 0.10% | 260 | 25 | 96 | 40.4 | 79.2 | White |
| 6 | 99% | 33% tg. | 0.10% | 240 | 120 | 92 | 21.8 | 40.2 | Beige |
| Effect of initiator content | | | | | | | | | |
| 7 | 99% | 33% tg. | 0.20% | 260 | 20 | 96 | 26.9 | 50.9 | White |
| 8 | 99% | 33% tg. | 0.30% | 260 | 10 | 95 | 15.0 | 25.1 | White |
| Effect of cyOEF purity | | | | | | | | | |
| 9 | 92% | 33% tg. | 0.10% | 280 | 4 | 95 | 14.2 | 30.3 | Brown |
| 10 | 88% | 33% tg. | 0.10% | 280 | 5 | 95 | 9.3 | 20.3 | Brown |
| Commercial PET bottle for comparison | | | | | | | | | |
| 11 | – | – | – | – | – | – | 31.5 | 60.9 | – |

$M_n$ and $M_w$ are absolute molecular weights from SEC coupled with multi-angle laser light scattering. Initiator was cyclic stannoxane in all cases. Polymer melt plasticisation was using a reacting polymer melt of twice the mass of the added cyOEF
*tg.* tetraglyme, *Conv.* conversion as SEC area of polymer over total area

beneficial compromise between slower reaction and more limited degradation, yielding the highest absolute molecular weights[24] without discolouration (Table 1, entries 4–6). A similar trade-off is observed for the impact of the initiator amount on the reaction: the PEF molecular weight in all cases builds in a living-like evolution, where reduced initiator amounts deliver higher molecular weights, as expected for ROP. However, this effect is counterbalanced by the slowed reaction due to the reduced number of catalytic sites, which consequentially leads to extended exposure to high temperature and degradation at longer reaction times (Fig. 3d, Table 1, entries 5, 7 and 8, Supplementary Table 2). Finally, cyOEF purities close to 99% enable high absolute molecular weight products, while lower purities result in lower molecular weights due to the impact of linear chain impurities acting as chain transfer agents (Table 1, entries 9 and 10).

## Discussion

Under optimised conditions, as presented in Fig. 3c, d and Table 1, ROP delivers bottle-grade (>95% conversion, $M_n$ > 30 kg mol$^{-1}$, uncoloured) PEF within less than 30 min, by which the usually observed degradation and discolouration can be avoided. In terms of material properties relevant for commercial applications, ROP-derived PEF, similar as polycondensation-derived PEF, exhibits superiority to bottle-grade PET (Table 1, entry 11): a higher glass transition (85 vs. 73 °C) provides better ambient thermal stability and a lower melting point (220 vs. 260 °C) reduces energy demand in existing post-processing steps. Superior mechanical strength complements the properties of PEF: about 50% higher tensile strength (76 vs. 50 MPa) and 70% higher Young's modulus (1894 vs. 1102 MPa) compared with bottle-grade PET make PEF a mechanically more resilient material in its final applications (Supplementary Table 4). The native PEF material, in agreement with literature values for polycondensation-based PEF, appears non-ductile with minor elongation at break (5 vs. 388% for PET)[9,25]. This is expected, since ductility is usually a result of the careful selection of suitable additives required to stabilise the polymer during post-processing[26]. In fact, molecular weight reduction below bottle-grade levels, as also reported for non-stabilised PET polyester, was observed during compression moulding (240 °C for several

minutes) for PEF tensile analysis after ROP synthesis. Typical polymer additives to be investigated may include antioxidants to mediate radical formation through shear, complexing agents to deactivate the residual metal catalysts and co-monomers to modify crystallisation behaviour. Nevertheless, an improved gas diffusion barrier (5× higher for an $M_n$ = 25 kg mol$^{-1}$ solution cast PEF film) improves shelf life of food and beverages in PEF packaging, and can enable the replacement of multi-layer composite materials which traditionally serve the purpose of higher gas barrier, where PET was insufficient thus far.

These aspects confirm the synthesis of PEF via ROP as an excellent candidate towards the so-called green bottle. A preliminary comparison of the energy requirements of an integrated ROP process, where the main factor is heat supply to the solvent during cyclisation, with the process based on polycondensation, where the major cost is associated with high vacuum power over long reaction times, resulted in similar values, indicating that ROP can be an economically competitive route. This process will promote the transition from fossil- to renewable-origin plastics with lower carbon footprint in some of the currently largest polymer markets. The idea of removing reaction by-products in a low viscosity step prior to polymerisation presents an improved pathway for other polymerisation systems that are burdened with diffusion limitations, as well. Moreover, the establishment of synthesis via ROP enables precisely the controlled architecture of PEF copolymers through a living-like process, as opposed to random polycondensation.

## Methods

**Materials**. Dibutyltin oxide (Bu$_2$SnO, Merck, ≥98%), anhydrous ethylene glycol (EG, Sigma Aldrich, 99.8%), dimethyl 2,5-furandicarboxylate (MeFDCA, CM fine chemicals, 99%), 2-ethylhexanoic acid tin(II) (tin octoate, SnOct$_2$, Sigma Aldrich, 95%), 1-dodecanol (Acros Organics, 98%), trifluoroacetic acid (TFA, Fluorochem, 99%), potassium trifluoroacetate (K-TFAc, Aldrich, 98%), 2-methylnaphthalene (Sigma Aldrich, 95%; AlfaAesar, 97%), 1,2-dichlorobenzene (DCB, Sigma Aldrich, 99%), acetonitrile (ACN, Sigma Aldrich, ≥99.7%), *n*-hexane (Sigma Aldrich, ≥95%), toluene (Fluka, ≥99.7%), tetraethylene glycol dimethyl ether (tetraglyme, Sigma Aldrich, 99%), diethyl ether (Et$_2$O, Sigma Aldrich, ≥99.8%), dichloromethane (DCM, Fisher, 99.99%), hexafluoroisopropanol (HFIP, Fluorochem, 99.9%), tetrahydrofuran (THF, Merck, >99.8%), chloroform-*d* (CDCl$_3$, Armar Chemicals, ≥99.8%) and trifluoroacetic acid-*d* (TFA-*d*, Cambridge Isotope Laboratories, 99.5%) were used as received. The initiators, 1-dodecanol and tetraglyme were stored in a glove box under a nitrogen atmosphere. For benchmarking our PEF with bottle-grade PET, samples of the latter were taken from

commercially available PET plastic bottles. To evaluate the accuracy of polymer molecular weight analytics, PET and polymethylmethacrylate (PMMA) standards were received from PSS Polymer Standards Service, Germany.

**Analytics**. Proton nuclear magnetic resonance ($^1$H NMR; 300 and 400 MHz) spectra were recorded on Bruker Avance III spectrometers and were referenced against the residual solvent signal (Supplementary Fig. 1). Conversion ($X$) and absolute molecular weights of PET and PEF (as number average, $M_n$, and weight average, $M_w$, respectively) were determined by size exclusion chromatography coupled with multi-angle laser light scattering (SEC-MALS). An Agilent 1100 gel permeation chromatography (GPC)/SEC unit was used equipped with two PFG linear M columns (PSS) in series with an Agilent 1100 variable wavelength detector/ultraviolet (VWD/UV) detector operated at 293 nm, a DAWN HELEOS II MALS detector and an Optilab T-rEX RI detector (both from Wyatt Technology Europe). Samples were eluted at room temperature in HFIP with 0.02 M K-TFAc at 1 mL min$^{-1}$. Conversion was evaluated with PSS WinGPC Unichrom software as the fraction of polymer versus overall UV signal area. Absolute molecular weights were evaluated in Wyatt ASTRA software with $dn/dc$ values based on our analytical setup ($dn/dc$ (PEF) = 0.227 mL g$^{-1}$, $dn/dc$ (PET) = 0.249 mL g$^{-1}$). Further absolute molecular weights were obtained from diffusion ordered spectroscopy (DOSY) NMR measurements using 0.4 mg mL$^{-1}$ samples of PET and PEF in TFA-$d$, and compared against relative molecular weights obtained with PMMA standards in SEC (Supplementary Table 3). Experimental parameters for DOSY measurements are described elsewhere[24].

Analysis of cyOEF ring-size composition and the purity from residual linear species was assessed on an Agilent 1100 high-performance liquid chromatography (HPLC) with UV detector at 280 nm followed by an Agilent 1640 single quadrupole electrospray ionization time-of-flight mass spectrometer (MS). Linear and cyclic species were unambiguously identified via this HPLC-MS setup and cyOEF purity was evaluated as the HPLC area associated with the peaks of the cyclic oligomers versus the overall HPLC area. UV absorptivity of linear and cyclic species was assessed prior to these measurements to ensure unbiased evaluation, and was found equal for all species[27]. Samples were dissolved in 15% (v/v) HFIP/CHCl$_3$ and eluted over an Eclipse XD8-C18 column (150 × 4.6 mm, 3.5 μm pore size) with an ACN/H$_2$O gradient from 20/80 to 80/20 over 40 min at 1 mL min$^{-1}$. Formic acid 0.1% (v/v) was added as stabiliser to both the organic and aqueous phases, using Millipore water as the aqueous phase and ACN as the organic phase. The injection volume was kept constant at 10 μL.

Thermal properties analysis was performed on a Mettler Toledo DSC Polymer machine calibrated with indium and zinc standards. The heating rate was 10 °C min$^{-1}$ under nitrogen flow at 50 mL min$^{-1}$. Cyclic oligomer melting points were derived from the first heating curve; glass transition temperature ($T_g$) and the melting point ($T_m$) of PEF were derived from the second heating curve after quenching in liquid nitrogen. $T_g$ was recorded at the midpoint temperature. Thermal stability and degradation were analysed on a Mettler Toledo TGA/SDTA 851 using a heating rate of 10 °C min$^{-1}$ under nitrogen flow of 50 mL min$^{-1}$. Degradation temperature was evaluated at 5% weight loss.

Polarised optical microscopy images were acquired with an Olympus BX51 microscope equipped with a Linkam LTS350 temperature controlled stage and a DP72 digital camera.

X-ray powder diffraction patterns were recorded on a Stoe&Cie STADI P Powder diffractometer with Cu-K alpha1 radiation, focusing Ge-Monochromator and Dectris Mythen Silicon Strip Detector.

Gas permeability analysis of ROP-based PEF and reference PET films was performed at 23 °C and 50% relative humidity on a MOCON Ox-Tran device using films of 35–45 μm thickness as obtained from solution casting. Film preparation was done by casting of about 200 mg mL$^{-1}$ PEF solutions in HFIP over a glass plate heated at 70 °C to evaporate HFIP, and film thickness was measured afterwards.

Tensile testing was performed on an Instron 5864 using dogbone specimen cut from PEF and PET films of known thickness at strain rates of 1 and 10 mm min$^{-1}$ at ambient temperature. Repetitions per sample were at least 3 per strain rate. The films were prepared by compression moulding of PEF from ROP, which had been vacuum-dried at 130 °C for several days and cryo-milled, and reference bottle-grade PET pellets, also vacuum-dried, at 230–240 °C for 3–6 min at 5 kN pressure.

**Preparation of cyclic stannoxane initiator**. 1,1,6,6-tetra-n-butyl-1,6-distanna-2,5,7,10-tetraoxacyclodecane, or commonly named cyclic stannoxane, was synthesised as previously reported[28,29]. In the glove box, dibutyltin oxide (4.06 g, 16.3 mmol, 1 eq.) was added to a two-neck round-bottom flask equipped with a magnetic stirrer. The flask was fitted with a cap and a Dean-and-Stark-trap equipped with a condenser. Toluene (105 mL) and anhydrous ethylene glycol (0.9 mL, 16.1 mmol, 1 eq.) were added and the mixture was heated to 145 °C. The solution was refluxed overnight, after which water (0.2 mL) had collected in the trap. Upon cooling the solution to room temperature, a white precipitate formed. The suspension was concentrated in vacuo to give a slurry and n-hexane (5 mL) was added at 0 °C. The white solid was collected by filtration and dried under vacuum at 45 °C for 2 h, yielding a white powder. $^1$H NMR (300 MHz, 25 °C, chloroform-$d$) δ (ppm) = 3.63 (t, $J$ = 15.5 Hz, 4 H, -CH$_2$-O-), 1.74 – 1.52 (m, 4 H, -CH$_2$-), 1.47–1.19 (m, 8 H, -CH$_2$-CH$_2$-), 0.92 (t, $J$ = 7.3 Hz, 6 H, -CH$_3$).

**Preparation of cyclic oligoethylene furanoate**. The cyOEF were prepared following the cyclisation protocols of high dilution methods described elsewhere[18,27,30]. For the route via depolymerisation in particular, MeFDCA (400 g, 2.17 mol, 1. eq), EG (270 g, 4.35 mol, 2 eq.) and Bu$_2$SnO (2 g, 8 mmol, 0.004 eq.) were charged in a three-neck round-bottom flask equipped with a magnetic stirrer under N$_2$ atmosphere. The solution was stirred at 180 °C and methanol as condensation product was distilled and removed over the course of 1–2 h. A high boiling solvent such as 2-methylnaphthalene or DCB was then added to dilute the obtained prepolymer to 10–30 g L$^{-1}$. The solution was stirred for 6–8 h at 180–200 °C and the cyOEF products were collected and purified from linear species via selective cooling precipitation and sometimes addition of 1:1 v/v hexane, followed by filtration. Further purification from linear species was achieved by adsorption either via elution in DCM with up to 7.5% v/v Et$_2$O over short silica gel columns, or via stirring of silica gel (1:1 by weight of MeFDCA) in DCB under reflux at 180 °C, if necessary. The product was concentrated, yielding the cycles as a white powder at purities of >99%. Individual cyclic species could also be fractionated by eluting crude product in DCM over silica gel using Et$_2$O/DCM (7.5/92.5 v/v) as mobile phase. All purification products were collected and their purity was analysed by HPLC-MS. $^1$H NMR (400 MHz, 25 °C, TFA-$d$) δ (ppm) = 7.55, 7.42 (s, 2 H), 4.9 (s, 4 H).

**Ring-opening polymerisation for the synthesis of polyethylene furanoate**. 300 mg cyOEF were weighed into a 5 mL Schlenk tube equipped with a magnetic stirrer in the glove box. The tube was transferred to a heating block and dried for about 1 h at 100 °C under vacuum. The Schlenk reactor was removed again and the vacuum was released with nitrogen, to add initiator at concentrations of 0.01 to 0.03 M in tetraglyme onto the solid cyOEF under N$_2$ flow. Initiator suspensions were preheated to 70 °C under stirring. The components in the Schlenk were mixed with a spatula. The reactor temperature was set to the desired value, and the Schlenk tube was reinserted into the heating block. In the case of initiator grinding, cyclic stannoxane initiator was ground into the cyOEF raw material in the glove box, and then transferred to the heat block in the 5 mL Schlenk tubes. In the case of polymer melt plasticisation reaction, cyOEF was polymerised initiated by injected initiator in tetraglyme for 10 min, after which additional cyOEF (50 and 100% by weight) were added. Samples for SEC analysis were collected along the reaction with spatulas, which had dried in the oven at 120 °C. The reaction was quenched in ice water and the product was dissolved in pure HFIP, then precipitated in THF and collected by decanting off the solvent and drying under vacuum at 80 °C, which yielded white or brown solids depending on the reaction conditions. $^1$H NMR (400 MHz, 25 °C, TFA-$d$) δ (ppm) = 7.46 (s, 2 H), 4.88 (s, 4 H).

**Data availability**. The authors declare that the data supporting the findings of this study are available within the article and its Supplementary Information file. All other relevant source data are available from the corresponding author upon reasonable request.

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

## Acknowledgements

The valuable work in the laboratory by our master students Laura De Lorenzi, Marcel Kröger, Edoardo Maggi, Fritz Polt and Harris Luk Ho Ting is gratefully acknowledged. We acknowledge funding through project 16120.1-PFEN-IW by the Swiss Commission for Technology and Innovation (CTI).

## Author contributions

All experiments were planned and analysed by J.-G.R. The synthesis of PEF was performed by J.-G.R. and D.K.H., and the synthesis of purified cyclic oligomers was performed by J.-G.R., D.K.H. and P.F. All work was supervised by G.S. and M.M. The manuscript was prepared by J.-G.R., while all authors contributed to refining of the latter in regular discussions.

## Additional information

**Competing interests:** The authors declare no competing interests.

