## [Peer Review File · Nature Communications]

Reviewers' comments:

Reviewer #1 (Remarks to the Author):

Polyethylene furanoate (PEF) currently represents a highly promising renewable resource-based bioplastic as replacement for fossil based polyethylene terephthalate (PET) with even improved material properties. It is indeed the focus of topical research both academically and industrially. The authors report the successful rapid (within minutes) synthesis of "green" bottle-grade high molar mass PEF ($M_n > 30$ kg/mol, conversions $> 95\%$) via ring-opening polymerization (ROP) from cyclic PEF oligomers (derived from 100% renewable-resource based monomers dimethyl furandicarboxylate and ethylene glycol) and "self-plasticizing" of the formed polymer itself. The approach enabled to avoid degradation and discoloration, major flaws of PET. The chemistry is sound and the polymers are suitably characterized (SEC, DSC, NMR, TGA and Polarised optical microscopy). Major achievements of this work also include an easy macromolecular concept and chemistry, a higher glass transition temperature and a melting temperature well above the degradation temperature of PEF, and an improved gas diffusion barrier, thus foreseeing valuable industrial processing and outcomes. Valuable comparison to commercial PET obtained from plastic bottle is also provided.

The mechanical properties of the PEF materials, again in comparison to alike PET, would nicely complement the results prior to publication.

Reviewer #2 (Remarks to the Author):

This paper shows that ring opening polymerization of cyclic oligomers of poly(ethylene furanoate) (PEF), in conjunction with added initiator and plasticizer can make low-defect, high molecular weight polymer. As PEF is an attractive alternative to poly(ethylene terephthalate) (PET), this has potentially very high impact for the bio-renewable polymer community and is thus very significant. My only issue with this paper is in the storytelling. The paper needs to be scrutinized to make sure everything is well-defined and described unambiguously. Some specific problems I found are below:

Line 8, define the units Mt/a

Line 23, "no ring-strain" should be "low ring-strain"

Figure 2a, Why are subscripted identifiers and non-subscripted identifiers used, apparently without rationale. I believe that one or the other should be used consistently

Figure 2a, Why is species C5 identified as two separate peaks?

Figure 2b, it is not defined as to what the three DSC traces are in this figure. Presumably different cyclic oligomers, but not stated

Figure 2d, the acronym POM is undefined.

Figure S5, experimental details for the experiments used to obtain these data and a definition of "signal ratio" (e.g. the y-axis of the plot) should be provided

Figure 3d, it is unclear what the arrow associated with plasticisation is trying to denote. The text indicates that addition of tetraglyme is the approach to introduce plasticisation, but the arrow is not pointing to those data.

Figure 3d, it is not indicated whether the initiator concentration is increasing or decreasing. Table 1 indicates what is happening, but the Figure caption should be re-written to make this easier to follow.

In general, figure captions should be revised to clearly indicate what data are being shown for each panel of the figure. Each panel should be described in turn. Current captions are a disjointed narrative that leave many things ambiguous or completely undefined.

Line 101, "A similar trade-off is observed for the impact of the initiator amount" should read "A similar trade-off is observed for the impact on the concentration of initiator" This noun string is repeated elsewhere.

Manuscript “Bottle-grade Polyethylene Furanoate from Ring-Opening Polymerisation of Cyclic Oligomers”

Response to reviewer comments

Reviewer #1 (Remarks to the Author):

Polyethylene furanoate (PEF) currently represents a highly promising renewable resource-based bioplastic as replacement for fossil based polyethylene terephthalate (PET) with even improved material properties. It is indeed the focus of topical research both academically and industrially.

The authors report the successful rapid (within minutes) synthesis of “green” bottle-grade high molar mass PEF ($M_n > 30$ kg/mol, conversions $> 95\%$) via ring-opening polymerization (ROP) from cyclic PEF oligomers (derived from 100% renewable-resource based monomers dimethyl furandicarboxylate and ethylene glycol) and “self-plasticizing” of the formed polymer itself. The approach enabled to avoid degradation and discoloration, major flaws of PET. The chemistry in sound and the polymers are suitably characterized (SEC, DSC, NMR, TGA and Polarised optical microscopy). Major achievements of this work also include an easy macromolecular concept and chemistry, a higher glass transition temperature and a melting temperature well above the degradation temperature of PEF, and an improved gas diffusion barrier, thus foreseeing valuable industrial processing and outcomes. Valuable comparison to commercial PET obtained from plastic bottle is also provided.

The mechanical properties of the PEF materials, again in comparison to alike PET, would nicely complement the results prior to publication.

Response:

We would like to thank the reviewer for the positive feedback, and have addressed his comment on mechanical properties, and tested our PEF product from ring-opening polymerisation against commercially available PET. We have successfully analysed the tensile strength and modulus of our PEF products, which lie in the expected range of polycondensation-based PEF as reported in the literature, and outperform commercial PET. In terms of strain at break, our material shows also the same brittle behaviour and little elongation as polycondensation-PEF, while ductility (larger strain at break) is a desired feature for commercial blow-molding applications. This, however, is a known issue in polymer processing: after synthesis of the native polymer, additives are usually applied to stabilise the material against the known degradation mechanisms (shear induced radical formation, residual active transesterification catalyst, hydrolysis, insufficient crystallization, etc.). We investigated the stabilisation effect of a variety of typically applied polyester additives, however, to an insufficient extent for the desired ductility. This screening would require larger amounts of polymer and work. While those results can be interesting for the industrial community, we believe that they do not provide a significant contribution to the herein presented results on the improved synthesis of PEF via ring-opening polymerisation. The results from the requested tensile property analysis are now discussed in the second-last paragraph in the manuscript and displayed quantitatively in Table S4.

Reviewer #2 (Remarks to the Author):

This paper shows that ring opening polymerization of cyclic oligomers of poly(ethylene furanoate) (PEF), in

conjunction with added initiator and plasticizer can make low-defect, high molecular weight polymer. As PEF is an attractive alternative to poly(ethylene terephthalate) (PET), this has potentially very high impact for the bio-renewable polymer community and is thus very significant. My only issue with this paper is in the storytelling. The paper needs to be scrutinized to make sure everything is well-defined and described unambiguously. Some specific problems I found are below:

We are grateful also for the kind evaluation of this reviewer and appreciate his comments to improve the comprehensibility of the manuscript. We addressed the reviewer's comments as follows:

Line 8, define the units Mt/a

Done.

Line 23, "no ring-strain' should be "low ring-strain"

Done.

Figure 2a, Why are subscripted identifiers and non-subscripted identifiers used, apparently without rationale. I believe that one or the other should be used consistently

The labelling has now been homogenised with non-subscripted identifiers.

Figure 2a, Why is species C5 identified as two separate peaks?

The error has been fixed and all peaks have been correctly labelled.

Figure 2b, it is not defined as to what the three DSC traces are in this figure. Presumably different cyclic oligomers, but not stated

In order to enhance clarification, the labels of the cyclic oligomers from Fig. 2a have been added also to Fig. 2b.

Figure 2d, the acronym POM is undefined.

POM is now defined.

Figure S5, experimental details for the experiments used to obtain these data and a definition of "signal ratio" (e.g. the y-axis of the plot) should be provided

The experimental conditions have been added to the plot.

Figure 3d, it is unclear what the arrow associated with plasticisation is trying to denote. The text indicates that addition of tetraglyme is the approach to introduce plasticisation, but the arrow is not pointing to those data.

The confusing arrow has been removed to give space for the actual data.

Figure 3d, it is not indicated whether the initiator concentration is increasing or decreasing. Table 1 indicates what is happening, but the Figure caption should be re-written to make this easier to follow.

The label on the arrow in Fig. 3d now reads "initiator increase".

In general, figure captions should be revised to clearly indicate what data are being shown for each panel of the figure. Each panel should be described in turn. Current captions are a disjointed narrative that leave many things ambiguous or completely undefined.

Figure captions have been rewritten accordingly.

Line 101, "A similar trade-off is observed for the impact of the initiator amount" should read "A similar trade-off is observed for the impact on the concentration of initiator" This noun string is repeated elsewhere.

The sentence has been rewritten for clarification.

REVIEWERS' COMMENTS:

Reviewer #1 (Remarks to the Author):

This present manuscript is the revised document of the initial manuscript entitled "Bottle-grade Polyethylene Furanoate from Ring-Opening Polymerisation of Cyclic Oligomers" by Prof Morbidelli and colleagues. The present manuscript has been revised in light of the referees' comments which have been addressed point by point in the response letter.

The various typos and acronyms to be defined have been fixed in the revised manuscript. Careful revision will enable to fix the last typos throughout the manuscript.

As suggested by the Referee, the authors have successfully analyzed the tensile strength and modulus of their PEF products, which fall in the expected range of polycondensation-based PEF, as reported in the literature; rewardingly, their PEF-products nicely outperform commercial PET. Regarding strain at break, the new PEF materials display also the same brittle behavior and small elongation as polycondensation-PEF, whereas ductility is desirable for commercial blow-molding applications. The authors have thus also preliminary investigated the stabilization effect of a variety of polyesters typically used as additives; however, the desired ductility could not be reached in these first attempts. While this screening did not provide a significant contribution to the presented results on the improved synthesis of PEF via ring-opening polymerization, it would indeed, as mentioned by the authors, require larger amounts of polymer and work, which subsequently would be the object of another manuscript. Accordingly, the tensile property analysis is discussed in the revised manuscript while data are displayed in ESI-Table S4.

Overall, the points raised by the referees in the previous round of review have been satisfactorily addressed. I thus support publication of the present manuscript in Nature Communications.